# Assessing COVID-19 Vaccine Booster Hesitancy Using the Modified 5C Scale in Zhejiang Province, China: A Cross-Sectional Study

**DOI:** 10.3390/vaccines11030706

**Published:** 2023-03-21

**Authors:** Xuan Deng, Yuchen Zhao, Shenyu Wang, Hanqing He, Zhiping Chen, Yang Zhou, Rui Yan, Xuewen Tang, Yao Zhu, Xiaoping Xu

**Affiliations:** 1Department of Immunization Program, Zhejiang Provincial Center for Disease Control and Prevention, Hangzhou 310051, China; 2Department of Preventive Medicine and Health Education, Fudan University, Shanghai 200032, China; 3Department of Epidemiology, Nanjing Medical University, Nanjing 211166, China

**Keywords:** COVID-19 vaccines, booster, vaccine hesitancy, China, 5C scale

## Abstract

Following the rollout of a booster campaign to promote immunity against COVID-19 in China, this study aimed to assess booster hesitancy among adults who were fully vaccinated with primary doses across Zhejiang Province. Firstly, the modified 5C scale developed by a German research team was assessed for reliability and validity via a pre-survey in Zhejiang Province. Then, a 30-item questionnaire was established to conduct online and offline surveys during 10 November to 15 December 2021. Demographic characteristics and information on previous vaccination experience, vaccine type of primary doses, attitudes towards booster doses and awareness of SARS-CoV-2 infection were collected. *Chi-square* tests, pairwise comparison and multivariate logistic regression were performed in data analysis. In total, 4039 valid questionnaires were analyzed, with booster hesitancy of 14.81%. Dissatisfaction with previous vaccination experience of primary doses (ORs = 1.771~8.025), less confidence in COVID-19 vaccines (OR = 3.511, 95%CI: 2.874~4.310), younger age compared to the elderly aged 51–60 years old (2.382, 1.274~4.545), lower education level (ORs = 1.707~2.100), weaker awareness of social responsibility of prevention and control of COVID-19 (1.587, 1.353~1.859), inconvenience of booster vaccination (1.539, 1.302~1.821), complacency regarding vaccine efficacy as well as self-health status (1.224, 1.056~1.415) and excessive trade-offs before vaccination (1.184, 1.005~1.398) were positively associated with booster hesitancy. Therefore, intelligent means should be strengthened to optimize vaccination services. More influential experts and other significant figures should be supported to promote timely evidence-based information via various media platforms to reduce public hesitancy and increase booster uptake.

## 1. Introduction

As of December 2022, over 650 million cases of severe acute respiratory syndrome coronavirus-2 (SARS-CoV-2) infection had been recorded, including over 6.65 million deaths globally [1]. The WHO declared the outbreak a Public Health Emergency of International Concern (PHEIC) on 30 January 2020 to accelerate the development of interventions in response to the pandemic [2]. So far, vaccines developed using various innovative technologies are regarded as the most effective public health strategy against severe illness and death caused by coronavirus disease (COVID-19) [3,4,5]. However, the increased transmissibility of the highly immune evasive variants, as well as the significant escape from immune protection elicited by current vaccines, have been widely reported [6,7], which has brought new challenges to the long-term fight. In response to the emerging global breakthrough infection, booster doses are believed to elicit longer-lasting and higher levels of protective antibody titers [8,9,10], which may help to control the outbreak and protect against serious diseases and death for people at high risk, in the absence of evidence-based effective treatment against COVID-19.

China began its initial vaccine campaign free of charge in September 2020 with a three-step strategy, firstly covering people at high risk, then expanding to the general adults, and lastly enrolling teenagers under 18 years old and children above 2 years old. As of October 2021, a total of 2.27 billion primary doses [11] had been administered, with a vaccination rate of 75.07% [12]. The majority were inactivated COVID-19 vaccines [13]. The detailed vaccination records can be tracked through mobile apps in China. Unfortunately, it seemed to be impossible to establish herd immunity only relying on current vaccines with the primary series [14]. To cope with the grave situation of imported outbreaks, China started a homologous booster campaign in October 2021. Following the updated clinical trials data on safety and efficacy [15,16,17,18,19,20], heterologous boosters based on inactivated primary doses were preferentially recommended in February 2022 due to superior immunogenicity [15,16,19,20]. Since China issue optimized COVID-19 control regulations on 7 December 2022 and prepared to reopen borders from 8 January 2023, the booster, especially for people at high risk, is one of the best protections to fight against consequent large-scale Omicron surge. Booster uptake highly relies on the willingness and acceptance of the public. Regrettably, the first booster rate was under 76% for people over 60 years old [21], the most group most vulnerable to Omicron infection in China, which will inevitably lead to immense pressure on medical institutions and even hospitalization or death. Given the tough situation, China has ramped up efforts to promote booster inoculation and recommend a second booster dose for certain groups at an interval of six months after the first booster since 14 December 2022 [22]. Four recombinant protein subunit vaccines and two viral vector vaccines against COVID-19 were newly approved for emergency use as boosters to meet diverse vaccine demand.

Globally, Israel [9], the United States [23], the United Kingdom [24], Chile [25], Japan [26] and Belgium [27] have also actively promoted booster campaigns since late July 2021. In addition, the second booster was approved in January 2022 in Israel and March 2022 in the United States, as studies indicated that vaccine efficacy of a second booster against overall infection, hospitalization and related deaths caused by Omicron was 34%, 64–67% and 72%, respectively, among the elderly aged 80 years [28]. Health authorities in Japan [29,30], Italy [31] and Australia [32] have even issued fifth doses of newly developed vaccines targeting Omicron variants to the elderly or the immunocompromised since July 2022. However, vaccine hesitancy (VH), which was noted as one of the 10 threats to global health in 2019 [33], seems to exist extensively all around the world, not only in countries with sufficient COVID-19 vaccine supplies but also in countries with a vaccine shortage [34]. Vaccine fatigue [35] was identified after several rounds of immunization demand against COVID-19 due to waning immunity, which significantly reduced trust in vaccine efficacy. Unfocused reports made the public underestimate the severity of Omicron variants. It is urgent for policymakers to explore the obstructions and underlying concerns of the booster, so as to encourage timely interventions to promote acceptance. Although previous studies [23,36,37,38,39,40,41,42,43,44,45,46,47] conducted similar surveys on booster hesitancy, the influential factors varied with different policy background, COVID-19 prevalence and predominant variants, COVID-19 vaccine type and supply capacity, public awareness of vaccine literacy and non-pharmaceutical interventions (NPIs), as well as investigation periods and religious beliefs.

So far, little is known about the attitudes towards the booster among the public in Zhejiang Province, a developed eastern coastal area of China which had frequent international trade and economic transactions before the pandemic. Thus, this cross-sectional study was primarily conducted to assess COVID-19 vaccine booster hesitancy among the adults in Zhejiang Province using a modified and validated 5C scale developed by a German research team [48]. The secondary objectives were: (1) to explore contributing factors of booster hesitancy to provide evidence-based advice for administrative decision-making; (2) to establish a validated 5C scale among the Chinese population to obtain accurate predictions for COVID-19 vaccine booster hesitancy.

## 2. Materials and Methods

### 2.1. Stage 1: 5C Scale Optimization and Localization

This phase aimed to establish a modified 5C scale on COVID-19 vaccine booster hesitancy among the Chinese population based on the original version developed by a German research team through interviews with immunization program experts and to test the reliability and validity of the modified version through pilot surveys among convenience samples in Zhejiang Province.

#### 2.1.1. Item Development

Based on the “3Cs” model proposed by the WHO EURO Vaccine Communications Working Group in 2011 [49], as well as the “Vaccine hesitancy determinants matrix” developed by The SAGE Working Group on Vaccine Hesitancy in 2014 [49], we optimized the validated 5C scale established by a German research team [48] in 2018 who took “*Calculation*” and “*Collective responsibility*” into account to further explain vaccination behavior from psychological perspectives, which was perfectly compatible with the globalization of the COVID-19 pandemic. In this scale, “*Calculation*” refers to individuals’ engagement in extensive information-searching prior to vaccination, such as evaluation of risks of COVID-19 infection, adverse reactions caused by vaccine and benefits of vaccination. “*Collective responsibility*” is defined as the willingness to protect others by one’s own vaccination to build herd immunity. Low values may indicate that a person is unaware of herd immunity or does not want to vaccinate for the benefit of others (especially those contraindicated to vaccination). The scale was adjusted from a 7-point to 5-point Likert-type (*Strongly agree* = 1, *agree* = 2, *not sure* = 3, *disagree* = 4 and *strongly disagree* = 5) to make it more concise. The intention to take COVID-19 vaccine boosters was measured by a 5-point Likert-type item (*Definitely will vaccinate =* 1, *likely to vaccinate =* 2, *not sure =* 3, *unlikely to vaccinate =* 4, *definitely will not vaccinate =* 5). We considered respondents who chose the latter three options (“3”, “4”, “5”) as “vaccine hesitators” according to the WHO definition of vaccine hesitancy; others were categorized as “vaccine acceptors” [49].

Six senior experts who majored in Immunization Programs from provincial, municipal and county-level centers for disease control and prevention (CDC), two public health physicians from community healthcare centers, one professor who majored in Mental Health and Psychiatric Epidemiology at Xiamen University and two translators working for an immunization program with a good bilingual background in both Chinese and English were interviewed face-to-face or online to evaluate the assessment dimension, cultural environment, content relevance and language expression and to make corresponding adjustments to ensure that the modified version was suitable for the actual situation in China. It should be noted that our goal was to explore the impact degree from five dimensions to explain vaccination behavior and to encourage targeted interventions for specific populations to decrease hesitancy. Thus, each dimension was treated as an influencing factor, rather than the total score of the whole 5C scale, due to uncertainty in weight allocation.

#### 2.1.2. Pilot Survey

In order to evaluate the reliability, validity and comprehensibility of the modified 5C scale, a pilot survey was conducted from 29 October to 3 November 2021 in Hangzhou city, the provincial capital of Zhejiang Province. At that time, only adults over 17 years old were approved for eligibility for the booster dose in China. Thus, the general public who had completed primary doses and had not yet received the booster was the target group for this booster hesitancy survey. Convenience samples were recruited, and the responses were anonymous, including adults, college students, university teachers and immunocompromised groups (HIV patients and cancer patients) from hospitals as well as people over 60 years old from communities. Considering network accessibility and education level, the first three types of participants were investigated through Wenjuanxing (WJX), a widely used online questionnaire survey platform, and the last two were interviewed face-to-face to ensure the survey was understood. Individual WJX accounts were limited to a single submission, and all the questions were compulsory. As compensation, participants who finished the questionnaire received a monetary incentive to improve cooperation. The sample size was determined by previous recommendations for sufficient power for scale construction [48] and the number of questions in the modified version scale.

#### 2.1.3. Reliability and Validity Assessment

Internal consistency reliability was assessed by calculating Cronbach’s α coefficient on both the whole scale and the 5-dimensional subscales. Considering recommendations from the Consensus-Based Standards for the Selection of Health Measurement Instruments (COSMIN) Manual, an alpha value ≥ 0.70 suggests good internal consistency reliability [48,50].

The Kaiser–Meyer–Olkin (KMO > 0.6) measure of sampling adequacy and Bartlett’s test of sphericity (*p* < 0.05) were conducted to ensure feasibility of factor analysis. Based on the exploratory factor analysis (EFA) conducted by a German research team [48], confirmatory factor analysis (CFA) was applied to further assess construct validity of the modified 5C scale used in our study, including the goodness of fit (GOF) for the overall scale structure, convergent validity (CV) and discriminant validity (DV) for the intra-dimensions of the subscales. Five model-fit indexes were calculated to evaluate GOF with suggested values in brackets: relative/normed chi-square $\left(\chi^{2}/df\leq 5\right)$, root mean square error of approximation (RMSEA < 0.08), non-normed fix index (NNFI > 0.9), incremental fit index (IFI > 0.9) and comparative fit index (CFI > 0.9) [51,52]. Three indexes were calculated to assess CV: standardized factor loadings (std. estimate > 0.5), average variance extracted (AVE > 0.5) and composite reliability (CR > 0.6) [53]. DV was assessed by comparison between the absolute value of inter-construct correlation coefficients (*r_i_*) and the corresponding square root of AVE [53] if the correlation was statistically significant (*p* < 0.05). DV was estimated to be good if |*r_i_*| < $\sqrt{AVE}$.

### 2.2. Stage 2: Investigating COVID-19 Vaccine Booster Hesitancy

This phase aimed to investigate the COVID-19 vaccine booster hesitancy among those who had completed primary doses and had not yet received the booster shot in Zhejiang Province. In total, 30 questions were included in the formal investigation to explore contributing factors and to provide evidence for targeted interventions to improve booster uptake, consisting of four parts: (1) sociodemographic characteristics; (2) previous vaccination experience and vaccine type of primary COVID-19 doses; (3) the validated and modified 5C scale established in Stage 1; (4) the attitudes towards booster doses and reasons for willingness or hesitancy regarding the booster. This study was deemed exempt by the Ethics Review Committee of Zhejiang Provincial Center for Disease Control and Prevention as no personal identifiers were collected in our questionnaire.

#### 2.2.1. Participants

Formal investigation was conducted from 10 November to 15 December 2021 in 4 cities (Hangzhou, Jiaxing, Taizhou and Quzhou City) in Zhejiang Province based on the economic development and COVID-19 risks. The source of participants, inclusion criteria and investigation method were consistent with pilot survey described in Stage 1. According to the risk of COVID-19 [54,55], medical workers and immunocompromised participants (those over 60 years old and people living with HIV/AIDS or various cancers) were defined as the high-risk group.

The sample size was calculated based on the hesitancy rate from the pilot survey to achieve 90% power for a 10% difference margin with the statistical parameters (*α* = 0.05) using the formula below:$$N=\frac{Z_{\alpha }^{2}\times P\times \left(1-P\right)}{\sigma^{2}}$$
where $Z_{\alpha }=1.96$, *P* = Hesitancy rate in pilot survey (18.35%) and *σ* = 0.1 × *P*. Thus, sample of at least 1893 participants should be recruited considering a loss-to-follow-up rate of 10%.

#### 2.2.2. Statistical Analysis

Descriptive analyses were used to evaluate the distributions in scale scores by vaccine hesitancy stages and individual characteristics. *Chi-square* tests were used to explore associated factors of booster hesitancy. The independent *t* test and analysis of variance were used to evaluate the differences in scale scores by sociodemographic characteristics, vaccine type, previous vaccination experience and vaccine hesitancy. A pairwise comparison was conducted using the *Bonferroni correction* method if the factor was significant. *Multivariate logistic regression* was performed to examine the comprehensive factors associated with booster hesitancy after controlling for sociodemographic confounders, vaccine type and previous experience of primary doses, as well as the five dimensions of the 5C scale. The *Hosmer–Lemeshow* test was used to test the goodness of fit. The collinearity test was carried out to assess the correlation between independent variables using a variance inflation factor (VIF) < 5. Data management was conducted using Microsoft Excel 365. Statistical analyses were conducted using IBM SPSS Version 26, IBM SPSS AMOS Version 26 and R Statistical Software Version 4.1.3. *p* ≤ 0.05 was considered statistically significant.

## 3. Results

### 3.1. Stage 1

#### 3.1.1. Overview

After two rounds of expert review and language translation, the modified 5C scale finally included 14 items (Appendix A) with five dimensions: *Confidence* (three items), *Complacency* (three items), *Constraint* (three items), *Calculation* (three items) and *Collective Responsibility* (two items). Each item was assigned scores of 1–5 in sequence according to the degree; for example, a score of 1 = strongly disagree, and a score of 5 = strongly agree. The score for each dimension equals the average score of items included. Three levels, “high”, “middle” and “low”, were defined as average score of the dimension >3, =3, and <3, respectively.

In total, 511 participants were investigated using the modified 5C scale, and 485 valid questionnaires (64.12% were 18–40 years old; 56.49% were male) were retrieved, with the response rate of 94.91%. A total of 89 participants demonstrated COVID-19 vaccine booster hesitancy (18.35%), of whom 68 were not sure (76.40%), 11 refused (12.36%) and 10 strongly refused (11.24%) to get a booster shot. Invalid questionnaires mainly resulted from missing items, mostly due to inability to understand the accurate meaning of “herd immunity” in the “*Collective Responsibility*” subscale.

#### 3.1.2. Reliability and Validity Assessment

##### Internal Consistency

Cronbach’s α coefficients for each subscale and the total modified scale were 0.903, 0.889, 0.897, 0.747, 0.665 and 0.817, indicating good scale reliability.

##### Construct Validity

The KMO measure of sampling adequacy and Bartlett’s test of sphericity for the scale were 0.838 and 3920.712 (*p* < 0.001), respectively, indicating that factor analysis was plausible.

Further GOF tests were conducted with the five model-fit indexes listed as follows: $\chi^{2}/df$ = 3.319, RMSEA = 0.066, NNFI = 0.952, IFI = 0.965 and CFL = 0.965, which indicated the modified 5C model fitted the collected data well.

Convergent validity was evaluated by three indexes list below in Table 1, reflecting highly related correlations between and within dimensions.

Similarly, discriminant validity was evaluated by comparison between the absolute value of inter-construct correlation coefficients and the corresponding square root of AVE, as listed in Table 2. According to the results, correlations between any two subscales were significant, and the coefficients were less than the square root of corresponding AVEs, indicating good DV between dimensions.

### 3.2. Stage 2

#### 3.2.1. Overview

In total, 4459 participants took part in the formal investigation using complete questionnaires (Appendix A), including 1273 offline samples (28.55%) and 3186 online samples (71.45%). In total, 4039 questionnaires were valid, with the response rate of 90.58%, 86.41% and 92.25% for the overall, offline, and online samples, respectively. Invalid questionnaires were due to incomplete information (233, 5.23%) and logical error (187, 4.19%). A summary of the sociodemographic, previous vaccination experience and corresponding vaccine type and the score distribution of 5C subscales of 4039 participants are provided in Table 3. In all, 598 participants (14.81%, 95%CI: 13.75–15.94%) demonstrated booster hesitancy, of whom 18 participants (3.01%) declared strong refusal, and 65 participants (10.87%) expressed negative attitudes towards booster vaccination. Univariate analysis of contributing factors indicated that age, educational status, annual income, marital status, vaccine type, previous vaccination experience of primary doses and all the five dimensions of the 5C scale were significantly associated with booster hesitancy. Further pairwise comparisons based on the *Bonferroni correction* method (Appendix A) revealed that a higher rate of hesitancy was observed among the elderly population (over 60 years old), lower educational level (middle school or below), lower annual income (RMB < 50,000), the widowed, college students, unemployed/retirees, those who received primary doses using inactivated COVID-19 vaccine (Sinopharm) and those who were unsatisfied with their previous vaccination experience. In addition, the five dimensions of the 5C scale were analyzed to further validate the application effect, and three levels were categorized according to the distribution of average score for each dimension (Table 3). Pairwise comparisons indicated that “*Confidence*” and “*Collective responsibility*” dimensions with higher score and “*Complacency*”, “*Constraint*” and “*Calculation*” dimensions with lower score indeed had significantly lower booster hesitancy rate, which met the expectations and indicated that the scale is a reliable tool to measure VH among the Chinese population.

#### 3.2.2. Influencing Factors in Modified 5C Scale

Significant associations were observed between individual characteristics and scores of subscales (Figure 1). Age, educational level, occupation and previous vaccination experience of primary COVID-19 doses were significantly correlated with all the five dimensions. Significantly higher scores were recorded for participants over 50 years old for “*Confident*”, “*Complacency*” and “*Constraint*” dimensions (*p* < 0.05), and lower scores for “*Calculation*” and “*Collective responsibility*” dimensions (*p* < 0.05). Respondents with a master’s degree showed higher scores for “*Complacency*” and “*Constraint*” dimensions (*p* < 0.01), and those with lower educational level (middle school and below) were less likely to “*Calculate*” and be “*Collective Responsible*” (*p* < 0.01) before getting the booster shot. Medical workers, civil servants and technical personnel and participants who had excellent or good previous vaccination experience of primary COVID-19 doses scored higher for “*Confident*” and “*Collective Responsibility*” dimensions (*p* < 0.05), and significantly lower for “*Constraint*” and “*Complacency*” dimensions (*p* < 0.05). Participants with mediocre experience tended to be less “*Calculative*” compared to those with a clear inclination (*p* < 0.05). As to other partial significant factors, male participants scored higher for “*Complacency*” and “*Constraint*” dimensions (*p* < 0.001). Urban participants had higher scores for “*Calculation*” and “*Collective responsibility*” dimensions (*p* < 0.01). Compared to married participants, single respondents scored higher for “*Calculation*” and “*Collective responsibility*” dimensions (*p* < 0.001). Lower annual income (<50,000 RMB) was associated with lower scores for “*Confidence*” and “*Collective responsibility*” dimensions (*p* < 0.001) and higher scores for “*Complacency*” and “*Constraint*” dimensions (*p* < 0.001). High-risk groups were less confident and thoughtful regarding COVID-19 vaccines when compared to the general population (*p* < 0.001).

#### 3.2.3. Influencing Factors for COVID-19 Vaccine Booster Hesitancy

A multivariate logistic regression model (Table 4) was established to explore influencing factors of COVID-19 vaccine booster hesitancy. The *Hosmer–Lemeshow* test was conducted and demonstrated a good fit of the logistic model with *χ^2^* = 4.609 (*p* = 0.798). No collinearity was detected among independent variables with VIF varying between 1.078 and 2.128. Age, education level, previous vaccination experience of primary COVID-19 doses and all the five subdimensions of the modified 5C scale were significant influencing factors associated with booster hesitancy (Table 4), among which previous vaccination experience had the greatest positive association (higher booster hesitancy) with ORs varying between 1.771 and 8.025, followed by the “*Constraint*” dimension (OR: 1.539, 95%CI: 1.302–1.821), “*Complacency*” dimension (1.224, 1.056–1.415) and “*Calculation*” dimension (1.184, 1.005–1.398). On the contrary, the “*Confidence*” dimension had the greatest negative association (lower booster hesitancy) with OR of 0.285 (0.232–0.348), followed by age of 51–60 years old (0.420, 0.220–0.785), educational level of master’s degree and above (0.476, 0.244–0.927), educational level of bachelor’s degree (0.586, 0.343–1.000) and the “*Collective Responsibility*” dimension (0.630, 0.538–0.739). The odds of COVID-19 booster hesitancy among participants who were not sure about the previous vaccination experience of primary COVID-19 doses was 8.025 (5.605–11.586) times the odds of booster hesitancy among participants who were very satisfied. With every point increase in “*Confident*” dimension score, the odds of being hesitant on the booster decreased by 0.285 (0.232–0.348). However, based on our model results, gender, residence, annual income, marital status, occupation and type of primary COVID-19 doses were not significantly associated with booster hesitancy.

## 4. Discussion

This cross-sectional survey in Zhejiang Province demonstrated the prevalence of COVID-19 vaccine booster hesitancy during November and December in 2021 using a modified and validated version of the 5C scale based on the “3Cs” model proposed by the WHO and the original 5C scale established by a German research team. A well-fitted multivariate logistic model was conducted to explore influencing factors of booster hesitancy among a large sample of 4039 participants from Zhejiang Province, China.

The modified five-point Likert 5C scale used in this study exhibits good reliability and validity, making it available and practical for measuring VH among the Chinese population. In total, 14 items were included, constituting five dimensions: “*Confidence*”, “*Complacency*”, “*Constraint*”, “*Calculation*” and “*Collective responsibility*”. For “*Confidence*” and “*Collective Responsibility*” dimensions, higher levels of agreement indicate lower booster hesitancy, whereas the relationship is reversed for the other three dimensions. With this scale, we found that the COVID-19 vaccine booster hesitancy was 14.81% in Zhejiang Province during November to December 2021, which was roughly consistent with the national investigation in China (16.1%) conducted in late January 2022 with an online sample of 898 [37]. Other two domestic web-based national surveys [36,41] carried out during November–December 2021 with thousands of samples demonstrated much higher booster acceptance rate (90.39%, 93.5%), which may be due to the different geographical region and epidemic background. It is likely that the hesitance will largely decline as China has started to ease COVID-19 restrictions since December 2022 due to decreasing pathogenicity of the Omicron variant, which may promote booster acceptance to cope with step-by-step reopening policy. Compared with similar studies abroad, the Japan community survey conducted in Fukushima (97.90%, 2439 samples) during September–October 2021 and the German survey conducted in December 2021 among university students and employees (87.80%, 930 samples), as well as the Italy community survey carried out in November–December 2021 based on the immunization center in Naples (85.70%, 615 samples) all demonstrated a positive attitude towards the booster in general. However, there is huge concern given that the booster hesitance rate reached 43.70% among 2647 Indonesian adults during December 2021 to January 2022 and 38.20% among 2138 adult Americans in July 2021, which was similar to the online survey conducted in Middle East and North Africa during November–December 2021 (39.80%, 3041 samples). Vaccine hesitancy has widely existed across the world to different degrees. Generally, our study enrolled a large sample of 4039 participants from all walks of life, covering residents from economically developed urban areas and underdeveloped rural areas using a combination of online and offline surveys, and the results provide some reference value. We believe that the Chinese respondents displayed a relatively high acceptance rate of the COVID-19 vaccine booster, which can be explained as follows: Firstly, the burden of COVID-19 in China is huge [56], both economically and psychologically. The Chinese people have a strong desire to move past the grief and sorrow brought by COVID-19. Secondly, China’s National Regulatory Authority (NRA) for vaccines has been assessed as meeting or exceeding all WHO standards in 2014 and achieved maturity level 3 in August 2022 [57], which means China has a stable, well-functioning and integrated regulatory system to ensure the quality, safety and effectiveness of vaccines that are manufactured, imported or distributed in the country. Furthermore, China was one of the first-line countries to develop COVID-19 vaccines via five different technical routes. Both the vaccines and the clinical treatments of SARS-CoV-2 infection are free for the public, regardless of health insurance, which indeed strongly increases people’s trust in government.

As a global phenomenon, booster hesitancy changes over time, and effective interventions can be carried out [49]. Compared with the online nationwide survey conducted in China [58,59,60] before the primary immunization campaign was launched, the prevalence of COVID-19 vaccine hesitancy was about 8.7–16.5% during March to May 2020, which was significantly associated with perceived benefits, efficacy, safety concerns of vaccines and vaccine price. However, during our survey period, the population involved had different levels of vaccine awareness based on previous experience of primary COVID-19 doses, which was most influential on people’s decisions regarding the booster from our multivariate logistic regression models. Among 4039 participants, 13.0% had unsatisfactory or bad experience with primary doses, and 78 participants gave specific reasons: 59% claimed to have adverse events after primary doses, 42.3% thought it took them a long time to get vaccinated, and 28.2% experienced noisy, unsanitary and muggy conditions at the vaccination site. At the beginning of the vaccine campaign, China indeed faced a severe vaccine shortage, leading to a vaccine rush. The vaccination sites were full of people once a wave of outbreaks occurred. However, so far, China has sufficient capacity for the booster, and the vaccination sites in Zhejiang Province provide daily services to meet vaccination needs. More intelligent information technologies are applied to allow easier and better-organized access to vaccination.

Our model indicated that most demographic characteristics were not significantly associated with booster hesitancy, including gender, residence, annual income, marital status, occupation and risk level of SARS-CoV-2 infection. However, young adults (18–30 years old) were much more vaccine-hesitant, which was consistent with studies in UK, USA and Japan [23,39,61]. In addition, lower education status was related to higher degree of hesitancy, which suggested that more interactive promotion methods and easy-to-understand models should be applied. Previous studies [62,63] indicate that social media plays an important role in health promoting, since it is fast, easy and highly convenient and efficient to spread information to Internet users. Treatment advice, psychological support and vaccine recommendations from clinical practitioners [38,42], such as doctors and nurses, were very important for encouraging the elderly to receive a booster, which necessitates a demand for vaccine literacy of elementary health care workers. Social media influencers as well as community leaders might be a better model to promote positive vaccine messages [38]. However, every coin has two sides, and misleading information on COVID-19 vaccines, especially about the adverse effects and contraindicators, should be detected promptly through the Internet and clarified with scientific data. The government and the scientific professors in epidemiology, immunology, clinical medicine and biomedical sciences have the responsibility to step forward to clarify false news before its spread gets out of control. A network public opinion monitoring system through AI-powered social media analytics would be helpful. The Chinese government needs to take advantage of multiple mediums to provide reliable and credible vaccine concepts and to correct misinformation in a timely manner. More scientific leaders, including scientific vloggers or bloggers, should be supported to continually spread updated vaccine knowledge and to resolve public confusion via social media, such as Weibo, WeChat, Bilibili, TikTok, Little Red Book and broadcast radio in China, which can reach people of different age groups and educational levels. To summarize, government media is essential, but promoting vaccine acceptance behavior can rely more on popular social media and Internet influencers from various fields so that health concepts can be better understood by the general public.

It is worth mentioning that the five dimensions from the modified 5C scale all provide significant insights into booster hesitancy, making them good psychological predictors. According to the adjusted ORs, the “*Confidence*” dimension has the strongest correlation, followed by “*Collective responsibility*”, “*Constraint*”, “*Complacency*” and “*Calculation*” dimensions. The first two had a positive association with booster acceptance, whereas the remaining three are negatively correlated. Compared to other studies [37,40,41,42,43] using different health belief models, such as the psychological drivers scale, including perceived severity, susceptibility, benefits, barriers and self-efficacy and cues to action; Vaccine Confidence Index; Vaccine Hesitancy Scale and 3Cs model from the WHO, our modified 5C scale comprehensively evaluated the booster hesitancy both from individual and collective perspectives. The detailed responses behind the modified 5C scale provide a direction for education content, which can inspire enthusiasm and interest among participants. Newly published studies [64,65] also demonstrated the positive relationship between trust levels and consumption of official sources of COVID-19 information and vaccine uptake, suggesting that communicating to the public early via various platforms with effective public health policies helps to achieve higher vaccine acceptance. Through our offline survey, we found that participants paid more attention to booster safety, vaccine indications and contraindications. Comparison between natural infection and vaccine-elicited immunity, including herd immunity, is also a popular topic. People seemed to care more about which type of booster vaccine or which immunization schedule could induce a higher level of protection. Moreover, where and when to get the booster is a topic of concern. Thus, dynamic public opinion monitoring via social media as well as offline investigation, combined with timely dissemination of credible and reliable booster information through experts from relevant fields, are needed to resolve public confusion and to effectively intervene in booster hesitancy.

Although this survey had a large sample with different demographic characteristics across Zhejiang Province through a validated 5C scale to comprehensively evaluate booster hesitancy, it had several limitations: Firstly, this is a cross-sectional survey using convenience sampling with a specific policy background, which could not reflect the dynamic trend of booster hesitancy and was unable to establish causality and generalizability between significantly associated factors and booster hesitancy. Secondly, to improve response rate, identifiable information was not collected, leading to inability to assess test–retest reliability for the modified 5C scale. Thirdly, some lifestyle habits, health behaviors and psychiatric background information, such as smoking and drinking frequency, washing hands, wearing masks, keeping social distance, knowledge and awareness of the COVID-19 pandemic and mental health status, were not measured or evaluated, and these factors might be potential confounders of booster hesitancy. Fourthly, selection bias and recall bias existed due to non-random sampling, non-response and self-reporting. Fifthly, we missed the opportunity to investigate vaccine hesitancy before the first implementation of COVID-19 primary doses in Zhejiang Province, which may provide clues to early targeted interventions through dynamic monitoring. Lastly, considering the COVID-19 pandemic situation in Zhejiang Province, we did not collect previous history of SARS-CoV-2 infection from participants, especially breakthrough infection, which impacted booster hesitancy in other studies [43]. However, as of 30 November 2021, the cumulative confirmed COVID-19 cases per million people was about 23.25 in Zhejiang Province [66,67], which was lower compared to the national prevalence (77.93 per 1,000,000) in China [12], not to mention the global prevalence (32,989.21 per 1,000,000) [12].

## 5. Conclusions

This study demonstrated that the COVID-19 vaccine booster hesitancy in Zhejiang Province during November–December 2021 was 14.81% among adults who had completed the primary series. Previous experience of primary doses had the strongest association with booster hesitancy from our multivariate logistic regression model, followed by the “*Confidence*” dimension, age, education level and the remaining four dimensions from the modified 5C scale. More intelligent technology should be applied in the process of vaccination services to avoid crowds and long waits. More influential scientific professors and experts from various fields as well as other significant figures (such as community leaders, influential bloggers and vloggers) should be supported to promote timely, credible and updated vaccine messages not only via traditional news media (such as TV, print media and radio) but also from network platforms (such as Weibo, WeChat, Bilibili and various social media tools), so as to comprehensively cover residents of all age groups and educational levels. Regular public opinion monitoring online and offline is a helpful means of increasing public awareness. To summarize, taking advantage of intelligent means to optimize vaccination services, gaining people’s confidence in the vaccine, making the public aware of the social responsibility of preventing and controlling COVID-19, reducing daily vaccination restrictions and underestimation of dangers of SARS-CoV-2 infection and relying more on experts and influential figures through various media platforms are essential in future public health interventions to increase booster acceptance as well as vaccine uptake.

## Figures and Tables

**Figure 1 vaccines-11-00706-f001:**
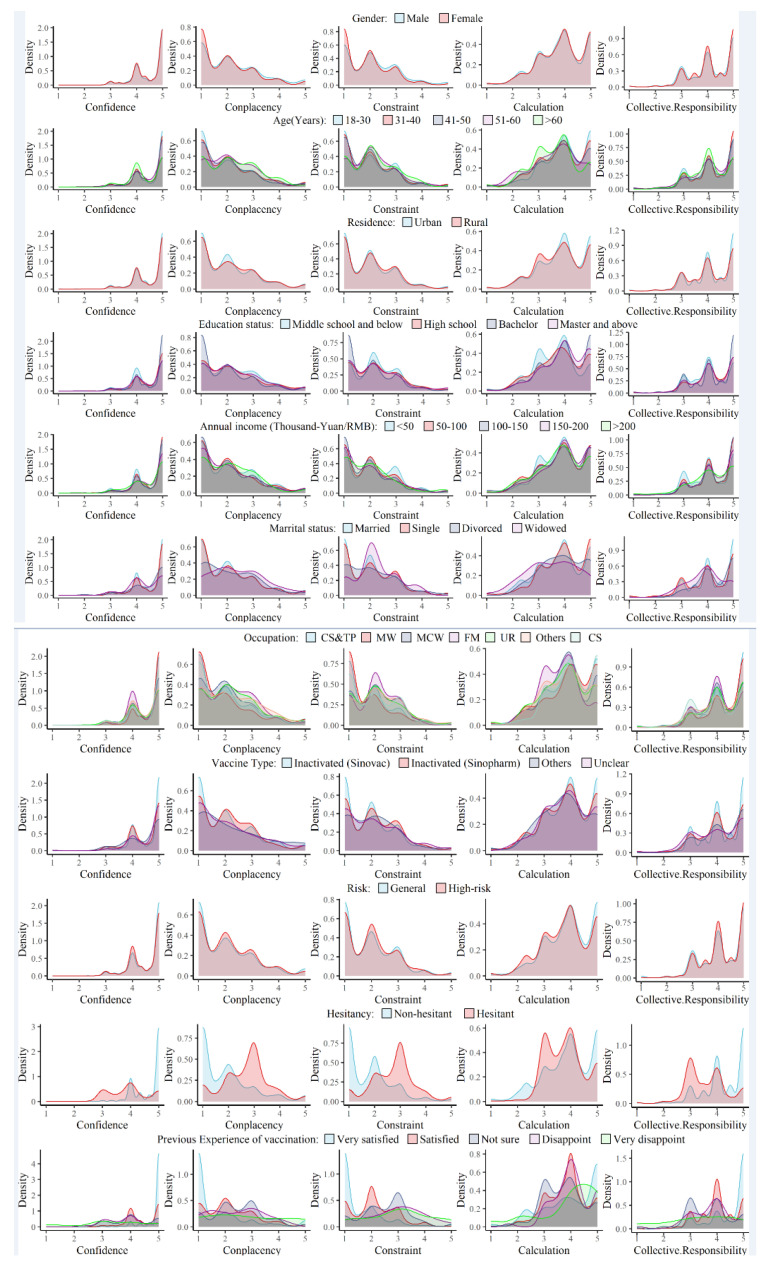
Probability density functions of each 5C subscale score in gender, age groups, residence, education status, annual income, marital status, occupation, risk of COVID-19, primary vaccine type and previous vaccination experience of primary doses of 4039 respondents. CS&TP: civil servant and technical personnel; MW: medical workers; MCW: manufacturing and commercial workers; FM: farmers/herders/fishermen; UR: unemployed/retirees; CS: college students.

**Table 1 vaccines-11-00706-t001:** Convergent validity evaluation of the modified 5C scale.

Dimension Items	*Std. Estimate*	*Reference*	*AVE*	*Reference*	*CR*	*Reference*
** *Confidence* **		0.5	0.769	0.5	0.909	0.6
1	0.797					
2	0.923					
3	0.906					
** *Complacency* **		0.5	0.734	0.5	0.892	0.6
1	0.888					
2	0.882					
3	0.796					
** *Constraint* **		0.5	0.753	0.5	0.901	0.6
1	0.795					
2	0.902					
3	0.902					
** *Calculation* **		0.5	0.534	0.5	0.768	0.6
1	0.546					
2	0.894					
3	0.711					
** *Collective responsibility* **		0.5	0.533	0.5	0.692	0.6
1	0.822					
2	0.625					

Note: *AVE* is the average variance extracted from the model; *CR* is composite reliability.

**Table 2 vaccines-11-00706-t002:** Correlation matrix of the modified 5C scale.

	*Confidence*		*Complacency*		*Constraint*		*Calculation*		*Collective Responsibility*
** *Complacency* **	0.173	*******							
** *Constraint* **	0.189	***	0.828	***					
** *Calculation* **	−0.101	*	0.201	***	0.249	***			
** *Collective responsibility* **	0.648	***	0.185	***	0.186	***	−0.304	***	
** *Square root of AVE* **	0.877		0.857		0.868		0.731		0.730

Note: *, *p* < 0.05; ***, *p* < 0.001. *AVE* is the average variance extracted from the model.

**Table 3 vaccines-11-00706-t003:** Sociodemographic, previous vaccination experience and vaccine type of primary COVID-19 doses and the score distribution of 5C subscales of 4039 participants.

Item	Total (*%*)	*No*. of Respondents with Booster Hesitancy (*%*)	*χ^2^*	*p*
** *Demographic characteristics* **				
**Gender**			1.276	0.259
Male	1819 (45.04)	282 (15.50)		
Female	2220 (54.96)	316 (14.23)		
**Age (Years)**			13.548	**0.009**
18–30	1698 (42.04)	274 (16.14)		
31–40	1072 (26.54)	133 (12.41)		
41–50	539 (13.34)	71 (13.17)		
51–60	247 (6.12)	32 (12.96)		
61–	483 (11.96)	88 (18.22)		
**Residence**			0.180	0.671
Urban	2576 (63.78)	386 (14.98)		
Rural	1463 (36.22)	212 (14.49)		
**Educational status**			11.615	**0.009**
Middle school and below	730 (18.07)	136 (18.63)		
Associate	406 (10.05)	60 (14.78)		
Bachelor	2490 (61.65)	338 (13.57)		
Master and above	413 (10.23)	64 (15.50)		
**Annual income (RMB)**			24.256	**<0.001**
<50,000	1629 (40.33)	289 (17.74)		
50,001–100,000	1149 (28.45)	165 (14.36)		
100,001–150,000	814 (20.15)	89 (10.93)		
150,001–200,000	290 (7.18)	32 (11.03)		
<200,000	157 (3.89)	23 (14.65)		
**Marital status**			159.026	<**0.001**
Married	2357 (58.36)	316 (13.41)		
Single	1550 (38.38)	259 (16.71)		
Divorced	80 (1.98)	13 (16.25)		
Widowed	52 (1.29)	40 (76.92)		
**Occupation**			45.407	<**0.001**
Civil servant and technical personnel	1249 (30.92)	149 (11.93)		
Medical worker	564 (13.96)	51 (9.04)		
Manufacturing and commercial worker	542 (13.42)	90 (16.61)		
Public service worker	58 (1.44)	10 (17.24)		
Farmer/herder/fisherman	324 (8.02)	49 (15.12)		
Unemployed/retiree	279 (6.91)	54 (19.35)		
College student	844 (20.9)	166 (19.67)		
Others	179 (4.43)	29 (16.20)		
**Risk of COVID-19**			2.207	0.137
High-risk	1869 (46.27)	260 (13.91)		
General	2170 (53.73)	338 (15.58)		
** *Vaccine Type* **				
**Brand of primary COVID-19 doses**			12.598	**0.027**
Inactivated (Sinovac)	2931 (72.57)	403 (13.75)		
Inactivated (Sinopharm)	868 (21.49)	159 (18.32)		
Viral vector	14 (0.35)	2 (14.29)		
Protein subunit	50 (1.24)	6 (12.00)		
mRNA	5 (0.12)	0 (0.00)		
Unclear	171 (4.23)	28 (16.37)		
** *Previous vaccination experience level of primary COVID-19 doses* **			800.347	<**0.001**
Very satisfied	1750 (43.33)	63 (3.60)		
Satisfied	1764 (43.67)	254 (14.40)		
Not sure	450 (11.14)	238 (52.89)		
Disappointed	58 (1.44)	32 (55.17)		
Very disappointed	17 (0.42)	11 (64.71)		
** *5C scale dimensions* **				
**Confidence**			547.266	<**0.001**
High *	3878 (96.01)	471 (12.15)		
Middle	145 (3.59)	116 (80.00)		
Low	16 (0.40)	11 (68.75)		
**Complacency**			376.195	<**0.001**
High	521 (12.90)	135 (25.91)		
Middle	393 (9.73)	170 (43.26)		
Low	3125 (77.37)	293 (9.38)		
**Constraint**			485.055	<**0.001**
High	357 (8.84)	142 (39.78)		
Middle	449 (11.12)	176 (39.20)		
Low	3233 (80.04)	280 (8.66)		
**Calculation**			100.956	<**0.001**
High	3110 (77)	441 (14.18)		
Middle	478 (11.83)	134 (28.03)		
Low	451 (11.17)	23 (5.10)		
**Collective responsibility**			312.95	<**0.001**
High	3366 (83.34)	352 (10.46)		
Middle	561 (13.89)	216 (38.50)		
Low	112 (2.77)	30 (26.79)		

Note: *: The “high”, “middle” and “low” level of each dimension in the modified 5C scale are defined as average score of the dimension >3, =3, and <3, respectively.

**Table 4 vaccines-11-00706-t004:** Associations between COVID-19 vaccine booster hesitancy and influencing factors from the multivariate logistic regression model.

Covariates	Estimate	Std. Error	*z* Value	*p* Value	OR	Lower Limit	Upper Limit
(Intercept)	3.225	0.700	4.610	**0.000**	25.155	6.419	99.716
**Gender**							
Male	1.00 (Ref)						
Female	−0.102	0.119	−0.858	0.391	0.903	0.715	1.140
**Age (Years old)**							
18–30	1.00 (Ref)						
31–40	−0.185	0.192	−0.962	0.336	0.831	0.570	1.213
41–50	−0.038	0.240	−0.157	0.875	0.963	0.600	1.541
51–60	−0.868	0.324	−2.681	**0.007**	0.420	0.220	0.785
61–	−0.319	0.317	−1.009	0.313	0.727	0.391	1.353
**Residence**							
Urban	1.00 (Ref)						
Rural	−0.163	0.127	−1.287	0.198	0.849	0.662	1.088
**Educational status**							
Middle school and below	1.00 (Ref)						
High school	−0.191	0.241	−0.792	0.428	0.826	0.514	1.322
Bachelor	−0.535	0.273	−1.961	**0.050**	0.586	0.343	1.000
Master and above	−0.742	0.341	−2.177	**0.029**	0.476	0.244	0.927
**Annual income (RMB)**							
<50,000	1.00 (Ref)						
50,001–100,000	0.234	0.180	1.301	0.193	1.264	0.889	1.799
100,001–150,000	0.004	0.225	0.020	0.984	1.004	0.646	1.560
150,001–200,000	−0.107	0.302	−0.354	0.723	0.898	0.492	1.611
>200,000	0.028	0.347	0.079	0.937	1.028	0.510	1.996
**Marital status**							
Married	1.00 (Ref)						
Single	0.013	0.196	0.065	0.948	1.013	0.688	1.487
Divorced	0.306	0.393	0.780	0.436	1.359	0.606	2.851
Widowed	−0.261	0.476	−0.549	0.583	0.770	0.289	1.886
**Occupation**							
Civil servant and technical personnel	1.00 (Ref)						
Medical worker	0.003	0.235	0.011	0.991	1.003	0.629	1.583
Manufacturing and commercial worker	0.049	0.222	0.222	0.824	1.050	0.679	1.619
Farmer/herder/fisherman	−0.148	0.319	−0.465	0.642	0.862	0.459	1.606
Unemployed/retiree	0.162	0.314	0.517	0.605	1.176	0.633	2.168
Others	0.009	0.275	0.031	0.975	1.009	0.583	1.720
College student	0.315	0.259	1.218	0.223	1.370	0.828	2.282
**Brand of primary COVID-19 doses**							
Inactivated (Sinovac)	1.00 (Ref)						
Inactivated (Sinopharm)	0.095	0.130	0.733	0.464	1.100	0.851	1.416
Others	−0.542	0.466	−1.163	0.245	0.582	0.219	1.376
Unclear	0.145	0.278	0.522	0.602	1.156	0.659	1.964
**Previous vaccination experience level of primary COVID-19 doses**							
Very satisfied	1.00 (Ref)						
Satisfied	0.571	0.162	3.529	**0.000**	1.771	1.295	2.446
Not sure	2.083	0.185	11.255	**<2 × 10^−16^**	8.025	5.605	11.586
Disappointed	1.774	0.362	4.895	**0.000**	5.897	2.893	12.017
Very disappointed	1.862	0.706	2.636	**0.008**	6.437	1.628	26.189
**Risk of COVID-19**							
High-risk	1.00 (Ref)						
General	−0.195	0.165	−1.179	0.238	0.823	0.595	1.138
** *Confidence* **	−1.256	0.104	−12.120	**<2 × 10^−16^**	0.285	0.232	0.348
** *Complacency* **	0.202	0.075	2.697	**0.007**	1.224	1.056	1.415
** *Constraint* **	0.431	0.086	5.043	**0.000**	1.539	1.302	1.821
** *Calculation* **	0.169	0.084	2.009	**0.045**	1.184	1.005	1.398
** *Collective Responsibility* **	−0.462	0.081	−5.681	**0.000**	0.630	0.538	0.739

## Data Availability

The data that support the findings of this study are available from the corresponding author upon reasonable request.

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
