# Peer review of "Assessing COVID-19 Vaccine Booster Hesitancy Using the Modified 5C Scale in Zhejiang Province, China: A Cross-Sectional Study"

_vaccines, 2023, doi:10.3390/vaccines11030706_

Round 1

Reviewer 1 Report

It is a very well written paper and it was a pleasure to read it. While elaborating on the population's hesitance toward vaccine boosters, the authors were able to reflect on and to refer to the international context. This certainly adds to the value of the paper. Methodology-wise: the study has been conducted with great care and the delivery is impeccable. The only thing that I would like the authors to consider is to add a few lines on booster's update in China in general; this data should exist. 

Author Response

Thank you very much for your nice comments. We are very encouraged. The updates on booster in China have been added in the paragraph 2 in the “Introduction” part (Line 63-71). Since no detailed data on the second booster was released in public by government, we just introduced the recent booster vaccine policy in general. Thanks again for your patience to review our manuscript.

Reviewer 2 Report

The manuscript is very long and not terribly exciting as is presented to encourage reading. If it were more concise in its presentation, it would increase its potential for attracting a wider audience. 

Corrections in the text have to be introduced, and I provide here a few examples.

Lines 53-54. Clarify would be clearer than clarity

Lines 54-55. The statement does not seem correct. Please, quote the reference.

Line 66. Should be groups instead of group.

Line 69. Should be pushed instead of push or change the verb.

Line 93. Should be a instead of an.

Line 374. Remove the word been.

Line 378. Change result to results.

Line 380. It said; China was the first country to suffer from COVID-19. There is no hint about COVID-19 originating in China. Something should be said  about this.

Line 381. Change has to have.

Lines 396-397. The paragraph needs to be reworded as it is not terribly clear.

Author Response

Reviewer2:

The manuscript is very long and not terribly exciting as is presented to encourage reading. If it were more concise in its presentation, it would increase its potential for attracting a wider audience. 

Corrections in the text have to be introduced, and I provide here a few examples.

Lines 53-54. Clarify would be clearer than clarity

Line 66. Should be groups instead of group.

Line 69. Should be pushed instead of push or change the verb.

Line 93. Should be a instead of an.

Line 374. Remove the word been.

Line 378. Change result to results.

Line 381. Change has to have.

Reply: We really feel sorry for the poor English language quality throughout the paper. We have carefully revised the manuscript according to your valuable comments and suggestions. In addition, we also ask for language editing service to make the whole article more rigorous, concise, and attractive. Thanks again for your patience to review our manuscript. Hope the modified version can meet your expectations.

Lines 54-55. The statement does not seem correct. Please, quote the reference.

Reply: Thanks for the comments. We have added some background on COVID-19 vaccine types as primary doses in China in the corresponding part to avoid confusions, as well as quoted more related references. We need to clarify that this paragraph is introducing COVID-19 vaccine policy in China, where over 90% recipients were administered with inactivated vaccine as primary doses in Zhejiang Province (according to unpublished government data). Clinical data on heterologous boosters (using recombinant protein vaccines or adenovirus vector vaccines as boosters) carried in China indicated superior immunogenicity compared with homologous booster (using inactivated vaccines as booster).

The articles listed below which have been quoted in the manuscript may provide some reference:

Li J, Hou L, Guo X, Jin P, Wu S, Zhu J, et al. Heterologous AD5-nCOV plus CoronaVac versus homologous CoronaVac vaccination: a randomized phase 4 trial. Nature medicine. 2022;28:401-9.

Li JX, Wu SP, Guo XL, Tang R, Huang BY, Chen XQ, et al. Safety and immunogenicity of heterologous boost immunisation with an orally administered aerosolised Ad5-nCoV after two-dose priming with an inactivated SARS-CoV-2 vaccine in Chinese adults: a randomised, open-label, single-centre trial. The Lancet Respiratory medicine. 2022;10:739-48.

Pérez-Then E, Lucas C, Monteiro VS, Miric M, Brache V, Cochon L, et al. Neutralizing antibodies against the SARS-CoV-2 Delta and Omicron variants following heterologous CoronaVac plus BNT162b2 booster vaccination. Nature medicine. 2022;28:481-5.

Zuo F, Abolhassani H, Du L, Piralla A, Bertoglio F, de Campos-Mata L, et al. Heterologous immunization with inactivated vaccine followed by mRNA-booster elicits strong immunity against SARS-CoV-2 Omicron variant. Nature communications. 2022;13:2670.

Wang Z, Zhao Z, Cui T, Huang M, Liu S, Su X, et al. Heterologous boosting with third dose of coronavirus disease recombinant subunit vaccine increases neutralizing antibodies and T cell immunity against different severe acute respiratory syndrome coronavirus 2 variants. Emerging microbes & infections. 2022;11:829-40.

Ai J, Zhang H, Zhang Q, Zhang Y, Lin K, Fu Z, et al. Recombinant protein subunit vaccine booster following two-dose inactivated vaccines dramatically enhanced anti-RBD responses and neutralizing titers against SARS-CoV-2 and Variants of Concern. Cell research. 2022;32:103-6.

Line 380. It said; China was the first country to suffer from COVID-19. There is no hint about COVID-19 originating in China. Something should be said about this.

Reply: Thanks very much for the comment. We do think it’s not appropriate to express the statement in that way. We have revised it in the corresponding part.

Lines 396-397. The paragraph needs to be reworded as it is not terribly clear.

Reply: Sorry for the confusions. We have revised the statement.

Reviewer 3 Report

This is an interesting paper that studies the association of vaccine/booster hesitancy with an adapted 5c scale of factors influencing vaccine hesitancy. The paper provides solid results and analysis, validation of the scale in the Chinese population, and novel aspects in the Zhejiang province in late 2021. The statistical analysis is well done, and the paper overall deserves to be published, however, some changes are required to make this more complete:

1) There are significant claims of causation throughout the paper, yet clearly all the results are an ASSOCIATION. Please do not say "XXX would significantly drive booster hesitancy". The current results are only associated. These sorts of comments are all over the paper and must be changed for the paper to be accepted.

2) The recruitment strategy, especially whether the 4000 individuals recruited are representative of the population in Zhejiang, and whether there are biases on the population must be admitted. 

3) A comparison between individuals recruited online vs offline must be added, and discuss any biases if found should be included

4) There needs to be a paragraph with all the limitations of the analysis and conclusion

5) Better connection with the literature needs to be done. This includes other papers published on Vaccines, especially on booster hesitancy (including populations outside China), for instance, these and other papers seem to be related to trust and consumption of information, so connections to such mechanisms should be discussed in the conclusion:

Latkin, Carl, et al. "Trusted information sources in the early months of the COVID-19 pandemic predict vaccination uptake over one year later." Vaccine 41.2 (2023): 573-580.

Juarez, Ruben, et al. "Dynamics of Trust and Consumption of COVID-19 Information Implicate a Mechanism for COVID-19 Vaccine and Booster Uptake." Vaccines 10.9 (2022): 1435.

6) Multiple sentences and assumptions make no sense or are difficult to understand. Correct English should be used. This is a non-exhaustive list:

- Abstract:  "Therefore, intelligent means should be strengthened to optimize vaccination service. More influential experts, even significant others, should be supported to promote timely evidence-based information via various media platforms, so as to reduce public hesitancy and increase booster uptake."

- Introduction: "it seemed to be impossible to establish herd immunity only rely on current vaccines with primary series"

- Introduction: "to get a better psychological prediction makes no sense"

- Several spacing need to be corrected throughout the paper, here's a non-inclusive list: Line 222: included14

Reviewer 4 Report

  • The authors could present the differences between the 5C scale from other scales about the COVID-19 vaccines.
  • I would like to see the differences between hesitancy in the first implementation and boosters of the COVID-19 vaccines.
  • The authors could also present the importance of this paper not only for Chinese readers but also for international ones.
  • The hesitancy about the COVID-19 vaccines could be related to mental disorders. So the authors could present the psychiatric background of the participants.
  • Although the sample seems adequate, the authors could present the results of their power analysis.
  • What do the authors mean by the risk of COVID-19? Did the authors mean healthcare professionals by high risk? Please explain with the references.
  • Figures (page 10 of 15) need help understanding. The authors could modify them.
  • Could the authors please explain the reason for taking age as a categorical variable? I kindly offer to take age as a continuous variable. Otherwise, the authors should discuss each category for age with references.
  • The hesitancy about COVID-19 vaccines could also be related to the kind of vaccine (mRNA, inactive). The authors could discuss this issue in more detail.

All in all, the authors could improve the manuscript after appropriate revisions.

Best regards

Round 2

Reviewer 2 Report

The manuscript has been improved by incorporating and/or correcting suggested changes.

Author Response

Thank you very much for your patience to review our manuscript. Your professional advice helps us a lot. Thanks again and hope everything goes well with you. 

Reviewer 3 Report

The author addressed my concerns. I recommend publication. 

Author Response

(The authors gave the same response as above.)
